# Vectorial Doppler metrology

Liang Fang[1,3], Zhenyu Wan[1,3], Andrew Forbes [1,2] & Jian Wang [1✉]

The Doppler effect is a universal wave phenomenon that has spurred a myriad of applications. In early manifestations, it was implemented by interference with a reference wave to infer linear velocities along the direction of motion, and more recently lateral and angular velocities using scalar phase structured light. A consequence of the scalar wave approach is that it is technically challenging to directly deduce the motion direction of moving targets. Here we overcome this challenge using vectorially structured light with spatially variant polarization, allowing the velocity and motion direction of a moving particle to be fully determined. Using what we call a vectorial Doppler effect, we conduct a proof of principle experiment and successfully measure the rotational velocity (magnitude and direction) of a moving isotropic particle. The instantaneous position of the moving particle is also tracked under the conditions of knowing its starting position and continuous tracking. Additionally, we discuss its applicability to anisotropic particle detection, and show its potential to distinguish the rotation and spin of the anisotropic particle and measure its rotational velocity and spin speed (magnitude and direction). Our demonstration opens the path to vectorial Doppler metrology for detection of universal motion vectors with vectorially structured light.

[1] Wuhan National Laboratory for Optoelectronics and School of Optical and Electronic Information, Huazhong University of Science and Technology, Wuhan, Hubei, China. [2] School of Physics, University of the Witwatersrand, Private Bag 3, Johannesburg, South Africa. [3]These author contributed equally: Liang Fang, Zhenyu Wan. ✉email: jwang@hust.edu.cn

The Doppler effect, discovered in 1842, is a universal wave phenomenon that has been widely applied to acoustic and optical metrology in astronomy, oceanography, medicine, engineering, etc.[1–19]. In particular, as for light wave, because of its ultra-high velocity, large bandwidth, and perfect directionality, the Doppler effect of light has spurred a myriad of applications, from cooling atoms to monitoring traffic flow. This effect originates from the relative motion between a wave source and an observer, resulting in a shift of the wave frequency. In early manifestations, the common linear Doppler effect was used for the deduction of linear velocities along the direction of wave (longitudinal motion). More recently, the rotational Doppler effect was revealed to allow the measurement of rotational velocities with helically phased acoustic and light waves[3–8].

Throughout the history of the Doppler effect, as well as its wide applications (e.g., Doppler metrology), it has always been based on scalar waves by detecting its blue or red shift of frequency. The Doppler shift is easy to detect by directly measuring scalar waves with low frequency, such as water and acoustic waves[8]. However, as for the light (electromagnetic) wave with ultra-high frequency (up to several hundreds of THz), it is not possible to detect the Doppler shift by directly measuring the frequency of the light wave with commercially available detectors. As an alternative, extracting the Doppler shift of light is usually implemented by interference with the coherent reference light. However, only the magnitude of motion velocity can be detected, while its direction is inaccessible unless additional techniques were adopted[20–24], for instance, the means of dual-frequency or heterodyne detection, using multiple laser beams, and a laser beam rapidly changing its direction of propagation, pointing at the object along different directions, as well as other schemes, with the help of a priori knowledge about the moving object under study and with the help of heavy computational efforts, but come at the cost of high complexity or practically infeasible in many real-world scenarios.

Apart from the well-established scalar (amplitude and phase) degrees of freedom, light also has a state of polarization (SoP) that describes the oscillated electric field in the plane perpendicular to the propagation direction. In particular, for structured light with tailored spatial polarization distribution, instead of the uniform SoP of conventional scalar optical fields, this new family of vectorial polarization fields (VPFs) is characterized by spatially variant polarized light fields across the transverse plane and may form polarization vortices (singularities)[25–28]. The most typical VPFs are azimuthal and radial polarization fields[25], and even higher-order VPFs, where the SoP can be well illustrated by the equator on the higher-order Poincaré sphere[29,30]. These cylindrical VPFs previously attracted a lot of attention as eigenmodes in optical fibers[31]. In the past decade, from fibers to free space and even to integrated devices, such fields have gained increasing interest and given rapid development in a diversity of applications in laser material processing[32], optical trapping[33,34], particle acceleration[35], classical entanglement[36–38], optical communications[39], quantum processing[40–42], microscopy and imaging[43,44], sensing and metrology[45], etc.

Here we reveal what we refer to as a vectorial Doppler effect that exploits such vectorially structured light fields. We exploit the polarization degree of freedom to extract directional information by following the Doppler response of the particle as it traverses a spatially structured polarization field. We outline the concept theoretically with the aid of well-known cylindrical VPFs and use this class of structured light to demonstrate proof of principle experiments for moving isotropic particles. We successfully measure the rotational velocity (magnitude and direction) of the moving particle. We also track the position of the moving particle under conditions of knowing the starting position and continuous tracking. The general case of a moving particle with anisotropy is also discussed. This may pave the way of vectorial Doppler metrology for complicated motions.

## Results

**Concept and principle.** The concept and operation principle of our vectorial Doppler effect with spatially variant polarized light fields are illustrated in Fig. 1, where a representative $HE_{31}$-like VPF is used for conceptual detection of the rotational or angular velocity of a moving particle. We first consider the simple case of an isotropic particle. When moving within a spatially variant polarized light field, it is assumed that the particle scatters the local light with the same polarization, similar to the micromirror reflection[15,16,24]. The reflected/scattered polarized light by the moving particle from this field can be interpreted as the Doppler polarization signal (DPS) with time-varying SoP. If the polarization distribution of this field is spatially periodic (for example, shown as Fig. 1a, c), the DPS can be written as a Jones vector

$$\mathbf{E}(v, t) \approx A \cdot \left[ \cos(k\gamma vt - \alpha) - \sigma \cdot \sin(k\gamma vt - \alpha) \right] \quad (1)$$

where $A$ is the real amplitude of the polarized light. Here we assume that the particle moves within a uniform intensity region in VPFs, i.e., giving a constant $A$. $k = 2\pi/\lambda$ is the wavenumber of light with $\lambda$ being the wavelength. $v$ denotes the linear velocity of the moving particle. $\gamma$ stands for the small angle between the Poynting vector of incident light and the observation direction, which determines the spatial period of polarization orientation ($\Lambda = \pi/k\gamma = \lambda/2\gamma$) of the VPFs. $\alpha$ is related to the initial polarization orientation (see Supplementary Notes 1 and 2). Note that the sign $\sigma$ takes "+1" or "−1", describing two opposite chirality of DPS, derived from the polarization orientation of $HE_{n,1}$ and $EH_{n,1}$ (fiber eigenmodes) like modes, respectively, where $n$ denotes the fold number of rotational symmetry.

If the moving particle reverses its velocity vector ($v \rightarrow -v$), the resulting DPS becomes

$$\mathbf{E}(-v, t) \approx A \cdot \left[ \cos(k\gamma vt + \alpha)\sigma \cdot \sin(k\gamma vt + \alpha) \right] \quad (2)$$

From Eqs. (1) and (2), one can see that no matter what values $\alpha$ takes, the DPSs always maintain $\mathbf{E}(-v, t) \neq \mathbf{E}(v, t)$. The DPSs show remarkable chirality inversion when reversing the motion direction of the particle, as shown in Fig. 1b, d. These two-dimensional DPSs belong to the distinct property of vectorial Doppler effect with spatially variant polarized light fields that enables the determination of the motion direction of a moving particle. By contrast, the classical Doppler effect based on scalar optical fields shows one-dimensional Doppler intensity signals, incapable of distinguishing the motion direction with a single scalar optical field (see Supplementary Note 1).

In practice, one can exploit the chirality of DPS to distinguish the motion direction by use of two polarizers. The phase relation of Doppler signals after passing through the polarizer with a polarizing angle $\theta_j$ (with respect to the $x$ axis) can be deduced as

$$I_j(v, t) = \left| \begin{pmatrix} 1 & 0 \\ 0 & 0 \end{pmatrix} \cdot \begin{pmatrix} \cos\theta_j & \sin\theta_j \\ -\sin\theta_j & \cos\theta_j \end{pmatrix} \cdot \mathbf{E}(v, t) \right|^2$$

$$\propto 1 + \cos[2(k\gamma vt + \sigma\theta_j - \alpha)] \quad (3)$$

where $j = 1, 2$ denotes polarizers 1 and 2 ($P_1$, $P_2$), respectively. It shows that the initial phase of this Doppler signal is dependent on the polarizing angle $\theta_j$ of the polarizer. For a single polarizer, obviously, one cannot distinguish the motion direction from the detected Doppler signal. This case actually falls into the traditional Doppler velocimetry with Doppler intensity signal based on a single scalar optical field (see Supplementary Note 1). If two polarizers ($P_1$ and $P_2$) are employed to detect the DPS

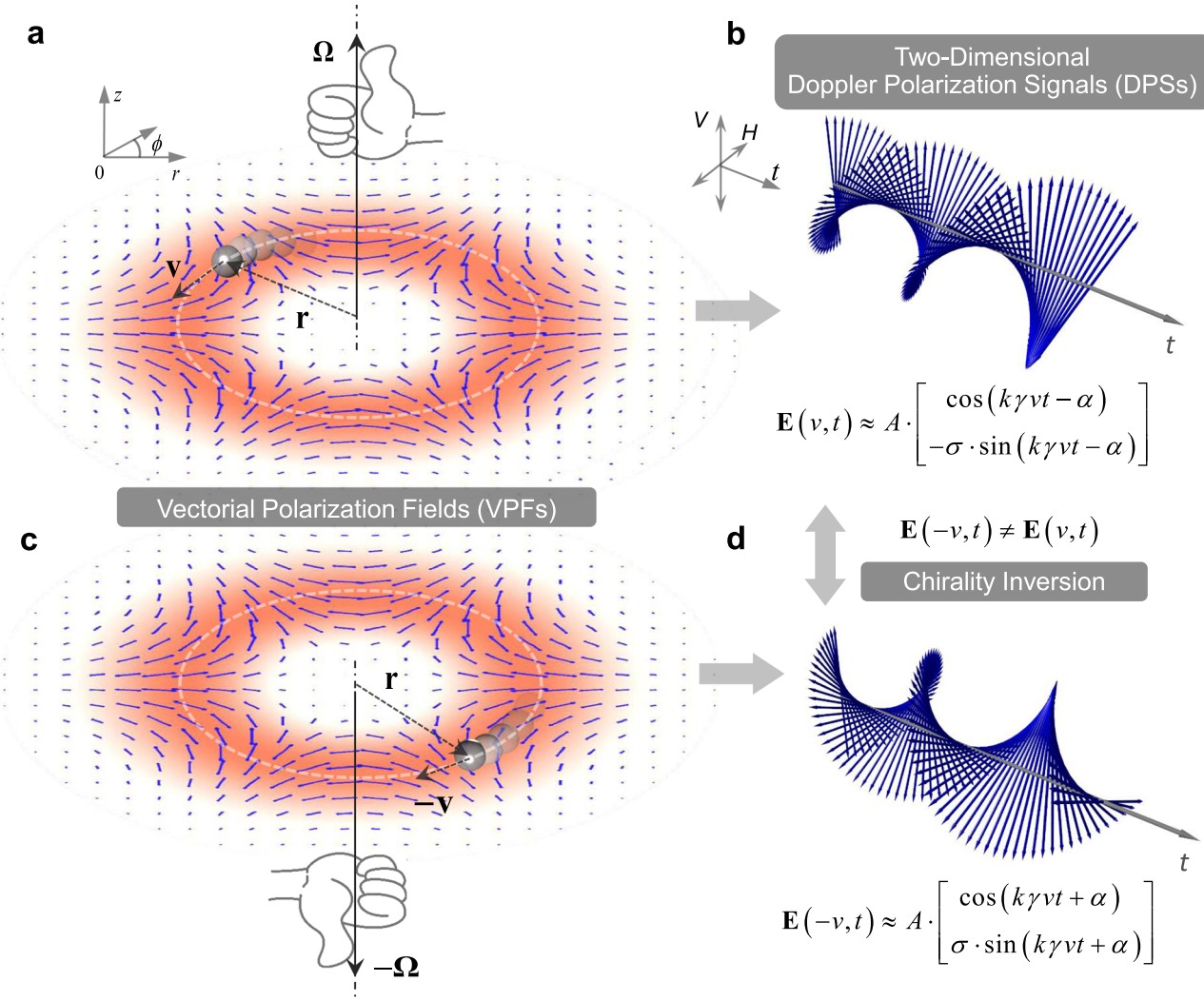

**Fig. 1 Vectorial Doppler effect with spatially variant polarized light fields used for determination of motion vector. a, c** The cylindrical $HE_{31}$-like field as a representative vectorial polarization field (VPF) is conceptually shown for direct detection of a moving particle with opposite rotational velocity vectors, respectively. **b, d** Two-dimensional Doppler polarization signals (DPSs) scattered from the VPFs show chirality inversion for opposite rotational velocity vectors. The two-dimensional DPSs carry full information (magnitude and direction) of velocity vectors, which is not accessible by classical Doppler effects based on a single scalar optical field. The blue arrows denote state of polarization (SoP). **r** radial vector, **v** linear velocity, **Ω** angular velocity.

synchronously, one can find a relative phase difference (RPD) between the detected Doppler intensity signals $I_1$ and $I_2$ after two polarizers, as shown in Fig. 2. From Eq. (3), the RPD between $I_1$ and $I_2$ can be written as $\Delta\varphi = \varphi_2 - \varphi_1 = \text{sign}(v) \cdot 2\sigma\Delta\theta$, where sign() stands for the sign function, and $\Delta\theta = \theta_2 - \theta_1$. It clearly shows that the RPD is dependent on the charity of DPS, as shown in Fig. 2a, b. Accordingly, the direction of the velocity vector can be determined by the measured RPD ($\Delta\varphi$), which can be defined as the Doppler phase shift. The magnitude of the velocity vector can be obtained from the common Doppler frequency shift $|\Delta f| = |k\gamma v/\pi|$. This inherent direction-sensitive vectorial Doppler effect benefits from the two-dimensional DPS of spatially variant polarized light fields. Furthermore, based on the principle of vectorial Doppler effect, significantly, one can also measure the velocity vector and track the position of the particle with variable motion, provided that the particle path and starting position are given and the particle is continuously tracked (see Methods).

Remarkably, for the traditional linear Doppler effect in longitudinal motion detection, the employed wave, treated as a whole, shifts its frequency because of the longitudinal

motion induced time-varying phase shift. For the vectorial Doppler effect in transverse motion (e.g., rotational motion) detection, the moving particle samples, reflects or scatters the local polarization of the high-order cylindrical VPF, giving a time-varying polarization. Additionally, since the employed high-order cylindrical VPF can be regarded as the superposition of two twisted-phase light components, the time-varying polarization of the scattered light is somehow equivalent to the time-varying phase shift. In general, not only the conventional Doppler velocimetry based on scalar optical field, but also the vectorial Doppler effect based on vectorially structured light, relies on modulating one of its physical quantities (i.e., phase, intensity, and polarization). This time-varying modulation onto the whole employed wave or the scattered part of the wave produces a frequency shift that is the Doppler shift.

**Experimental detection of rotational velocity vector.** We first demonstrate the detection of the rotational velocity vector of a rotating particle by the vectorial Doppler effect using a cylindrical VPF, as shown in Fig. 3a. In principle, for a particle that rotates

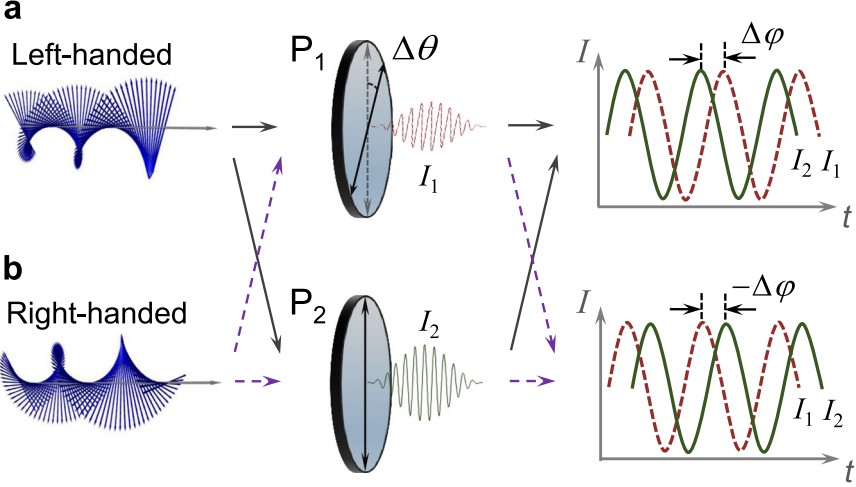

**Fig. 2 Two-dimensional DPSs can be detected by two polarizers to determine both the magnitude and direction of the motion vector. a** The left-handed DPS passing through two polarizers ($P_1$ and $P_2$) gives a fixed relative phase difference (RPD) between the detected Doppler intensity signals 1 and 2 ($I_1$ and $I_2$), known as Doppler phase shift. **b** The right-handed DPS passing through $P_1$ and $P_2$ gives an opposite Doppler phase shift between $I_1$ and $I_2$, relative to the former.

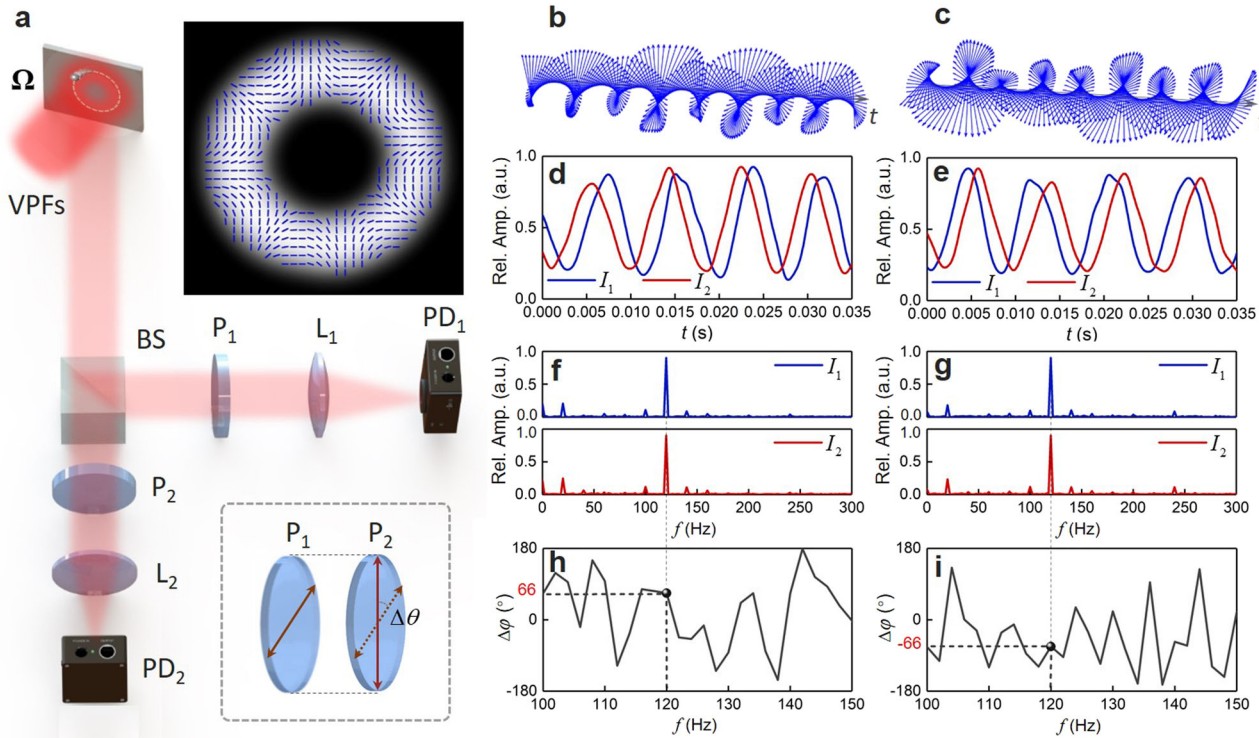

**Fig. 3 Experimental configuration and measured results. a** The cylindrical $HE_{41}$-like VPF illuminates a rotating particle, and the reflected DPS is split into two paths by a beam splitter (BS), then filtered by two polarizers (P1 and P2) with a polarizing angle difference of $\Delta\theta$ (down inset), and finally detected by two photodetectors (PD1 and PD2). The up inset shows the measured intensity and polarization distributions of the high-quality $HE_{41}$-like VPF generated in the experiment. L1, L2: lens. **b, c** Recovered DPSs show right-handed and left-handed under two opposite rotational velocities. **d, e** Measured Doppler intensity signals by $PD_1$ and $PD_2$, respectively. **f–i** Doppler frequency spectra acquired by fast Fourier transform (FFT) for the recorded Doppler intensity signals. **f, g** Amplitude–frequency spectra. **h, i** Phase-frequency spectra as RPDs between the recorded Doppler intensity signals 1 and 2. **b, d, f, h** $\Omega = 40\pi$ rad/s. **c, e, g, i** $\Omega = -40\pi$ rad/s. a.u. arbitrary unit.

around an axis with a linear velocity **v**, perpendicular to the radial vector **r**, the rotational or angular velocity vector (pseudovector) can be defined by $\Omega = (\mathbf{r} \times \mathbf{v})/r^2$, where $r$ is radius. The magnitude of the rotational velocity is determined by the rate of change of the angular position of the rotating particle, usually expressed in radians per second, i.e., $\Omega = d\phi/dt$. The direction ("+" or "−") of $\Omega$ is either up or down along the axis of rotation by the right-

hand rule, as shown in Fig. 1a, c. In general, the cylindrical VPF can be generated by many approaches, such as the superposition of two light components with different twisted phase structures $\pm\ell$ and circular polarization $\sigma = \pm 1$[29,30], where $\ell$ represents the topological charge number (mode order). In this case, the angle $\gamma = \ell/kr$ in Eqs. (1)–(3) represents the skew angle of the Poynting vector of the twisted light around the propagation

axis[46,47], and the magnitude of the linear velocity is $v = r\Omega$. Therefore, the DPS in Eq. (1) can be written as

$$\mathbf{E}(\Omega, t) \approx A \cdot [\cos(\ell\Omega t - \alpha) - \sigma \cdot \sin(\ell\Omega t - \alpha)] \quad (4)$$

In the proof of principle experiment, as shown in Fig. 3a, we generate a high-quality $HE_{41}$-like cylindrical VPF using a Sagnac interferometer configuration with an incorporated spatial light modulator (SLM) to detect the rotational velocity vector of a rotating particle that is mimicked by a digital micromirror device (DMD) (see Methods and Supplementary Note 4). The cylindrical VPF is properly adjusted to match the rotational trajectory of the particle when illuminating this rotating particle. The reflected DPSs shown in Fig. 3b, c by this rotating particle with two opposite rotational velocities are split into two paths, and then pass through polarizers 1 and 2 ($P_1$ and $P_2$) in each path with the polarizing angle difference of $\Delta\theta \approx 33°$. After that, the corresponding Doppler intensity signals 1 and 2 are collected by the photodetectors 1 and 2 ($PD_1$ and $PD_2$), as plotted in Fig. 3d, e, respectively. The Doppler frequency spectra in Fig. 3f–i are acquired by fast Fourier transform (FFT) for the Doppler intensity signals, including both amplitude–frequency spectra, as shown in Fig. 3f, g, as well as phase-frequency spectra, as shown in Fig. 3h, i. For convenience, we present the phase-frequency spectra plotted as the RPDs between the Doppler intensity signals 1 and 2. From the experimental results, one can clearly see that the DPSs reverse their chirality in Fig. 3b, c and thus give rise to the delay/advance in time between Doppler intensity signals 1 and 2 in Fig. 3d, e under opposite rotational velocities. All the measured amplitude–frequency spectra show a fixed Doppler frequency shift in Fig. 3f, g. However, the corresponding Doppler phase shifts give opposite values in Fig. 3h, i, i.e., $\Delta\varphi \approx 66°$ for $\Omega = 40\pi\,\mathrm{rad/s}$, while $\Delta\varphi \approx -66°$ for $\Omega = -40\pi\,\mathrm{rad/s}$, in accordance with the theories.

With the same twisted phase structure basis forming the $HE_{41}$-like VPF, the cylindrical $EH_{21}$-like VPF is also employed in the experiment. The comparison results show that the RPDs at the Doppler shift peaks reverse the symbols compared to the results using $HE_{41}$-like VPF (see Supplementary Fig. 6). We further show the measured results using cylindrical VPFs with different orders ($TM_{01}$, $HE_{21}$, $EH_{21}$, $HE_{41}$, $EH_{41}$, $HE_{61}$, $EH_{61}$, $HE_{81}$, $EH_{81}$, and $HE_{10,1}$), as plotted in Fig. 4a, as well as the measured results using $HE_{61}$-like VPF under different rotational velocities (see Supplementary Fig. 7). The experimental results verify that the mode order ($\ell$) of cylindrical VPFs and magnitude of rotational velocities have a similar relationship with the Doppler frequency shift, i.e., the larger the mode order of cylindrical VPFs and

magnitude of rotational velocities, the larger Doppler frequency shift obtained. Meanwhile, the form ($\sigma$) of cylindrical VPFs and the direction of rotational velocities also show a similar relationship with Doppler phase shift.

We also present the experimental results for Doppler phase shifts ($\Delta\varphi$) as a function of the polarizing angle difference ($\Delta\theta$) when detecting the DPSs using two polarizers, as shown in Fig. 4b. The Doppler phase shifts change twice as much as the polarizing angle difference and the sign of them also depends on the types of employed VPFs (HE- or EH-like VPFs), which is well consistent with theories ($\Delta\varphi = \mathrm{sign}(\Omega) \cdot 2\sigma\Delta\theta$). Note that the polarizing angle difference should be avoided around $\pm 90°$ in the experiment, since the resulting Doppler phase shifts of $\pm 180°$ and $\mp 180°$ for opposite velocity vectors are equivalent in Doppler phase-frequency spectra and not distinguishable. All the measured results verify that the spatially variant polarized light fields possess an inherently direction-sensitive (vectorial) Doppler effect that enables the determination of velocity vectors of moving targets.

**Experimental tracking of instantaneous position and velocity.** We further demonstrate the real-time tracking of instantaneous position and velocity of a moving particle with a complex motion vector based on vectorial Doppler effect with spatially variant polarized light fields. It is worth noting that the knowledge of particle path, starting position, and continuous tracking is the necessary condition for real-time instantaneous position tracking. Here we first use a higher-order cylindrical VPF to implement the real-time tracking of angular position and velocity of a representative pendulum motion. The particle mimicked by DMD moving in such a state rotates around the beam axis of a $HE_{19,1}$-like VPF, as shown in Fig. 5a. Despite, strictly speaking, not being a harmonic motion for the pendulum motion with a pivot angle, the moving particle here is approximately controlled as the state of harmonic motion. The pendulum length is set as about 2 mm in line with the radius of cylindrical VPF. In this case, the moving particle with harmonic motion will reflect the local polarized light from the higher-order cylindrical VPF to a nonuniform DPS, as shown in Fig. 5b. This DPS carries the instantaneous motion information of the pendulum motion. The density of time-varying polarization variation of DPS shows the magnitude variation of the velocity vector, while the chirality of DPS indicates the direction information, and especially, the chirality inversion implies the direction reverse of the movement.

In the experiment, after acquiring the DPS in Fig. 5b through two polarizers, we obtain the Doppler intensity signals in Fig. 5c.

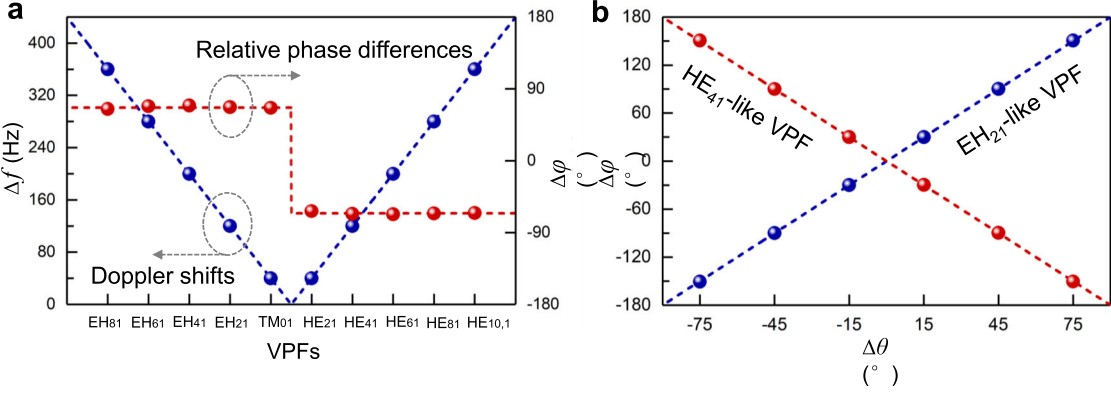

**Fig. 4 Measured results for vectorial Doppler effect using cylindrical VPFs with different orders. a** Doppler frequency shift and Doppler phase shift versus different orders of cylindrical VPFs ($TM_{01}$, $HE_{21}$, $EH_{21}$, $HE_{41}$, $EH_{41}$, $HE_{61}$, $EH_{61}$, $HE_{81}$, $EH_{81}$, and $HE_{10,1}$). **b** Doppler phase shift versus polarizing angle difference between two polarizers used to detect the DPSs. The angular velocity is $\Omega = -40\pi\,\mathrm{rad/s}$. The dots represent measured results and the dashed lines denote theories.

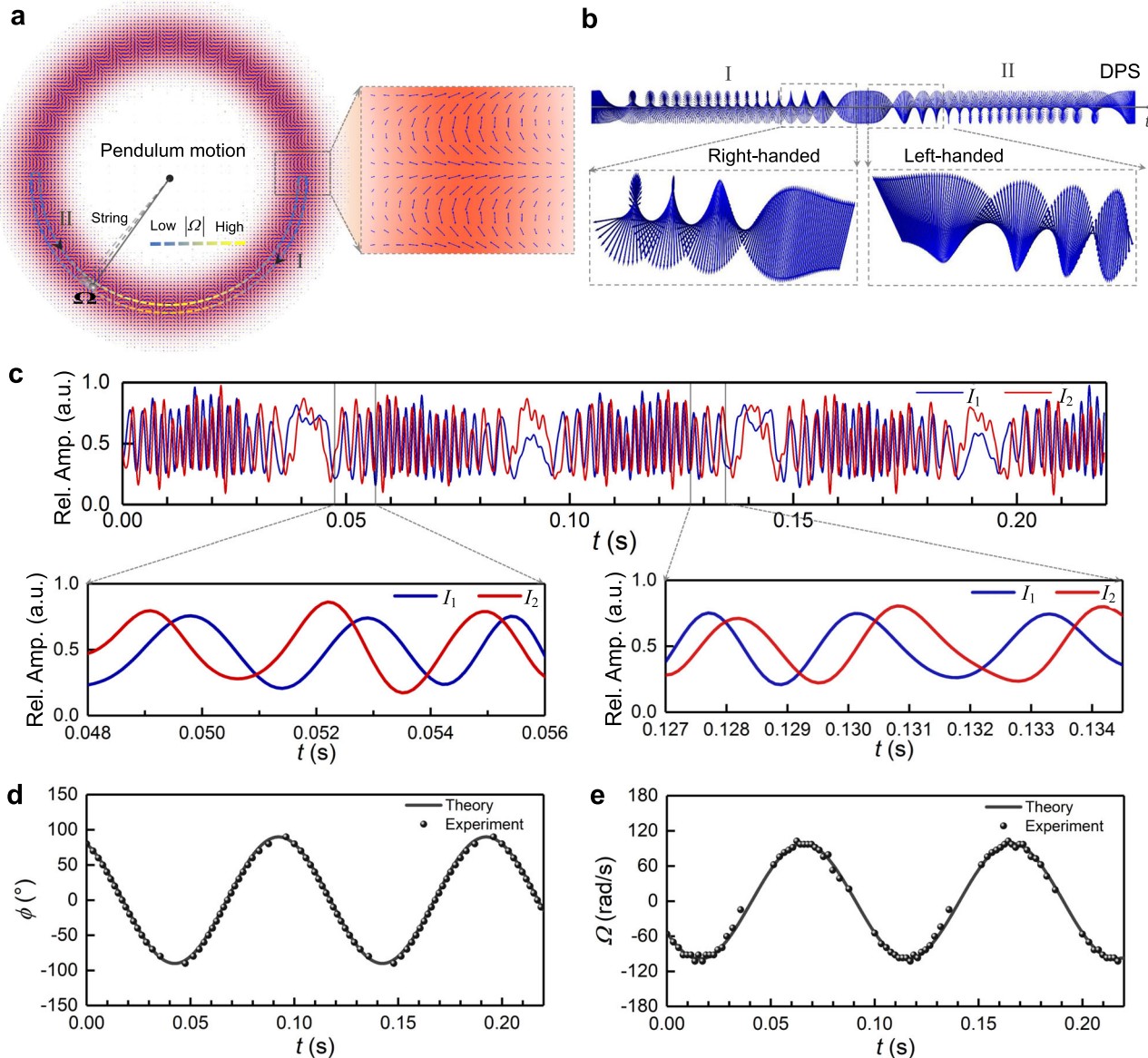

**Fig. 5 Real-time tracking of instantaneous position and velocity of a complicated moving particle based on vectorial Doppler effect with spatially variant polarized light fields, provided that the particle path and starting position of the moving particle are given and the particle is continuously tracked. a** A moving particle with pendulum motion is illuminated by the higher-order cylindrical $HE_{19,1}$-like VPF. The inset shows zoom-in details of spatial polarization distribution of the $HE_{19,1}$-like VPF. **b** DPS (Simulated) reflected from the $HE_{19,1}$-like VPF is available to retrieve the instantaneous position and velocity vector information of the moving particle. **c** Measured Doppler intensity signals after filtering the DPS (Fig. 5b) by two polarizers. The insets are zoom-in details of the measured Doppler intensity signals in two different time windows. a.u. arbitrary unit. **d** Retrieved instantaneous angular positions. **e** Retrieved instantaneous angular velocities. The dots represent experiments and the solid lines denote theories.

According to the principle of real-time tracking (see Methods), under conditions of the known starting position and continuous tracking, we successfully retrieve the instantaneous angular positions in Fig. 5d and angular velocities in Fig. 5e of the particle with a pendulum motion. More generally, we also experimentally demonstrate the real-time tracking of the moving particle with random rotation around the beam (see Supplementary Fig. 8). All the measured results verify the feasibility of real-time tracking of a complicated moving particle based on vectorial Doppler effect with spatially variant polarized light fields, provided that the particle path and starting position of the moving particle are given and the particle is continuously tracked.

In the real-time tracking of a particle with complicated movement, there will be inevitably less peaks in the Doppler

intensity signals from the spatially variant polarized light fields when the velocity is close to zero, resulting in fewer measured data points and thus reduced resolution, as shown in Fig. 5d, e. Even so, the temporal resolution of the real-time tracking can be further enhanced by exploiting even high-order cylindrical VPFs with reduced spatial period $\Lambda = \pi/k\gamma = \lambda/2\gamma$ of the spatially variant polarized light fields. In practical tracking applications, one can generally increase the observation angle $\gamma$ and use the laser with a shorter wavelength to improve the temporal resolution. Especially, for the cylindrical VPFs ($\gamma = \ell/kr$), it means that the temporal resolution can be improved by increasing the mode order ($\ell$) and reducing the size ($r$) of cylindrical VPFs. This is obvious as more peaks are available to indicate the motion variation in the Doppler intensity signals

with higher-order cylindrical VPFs, because of the subtler angular period of the spatial polarization distribution. Nonetheless, it is noteworthy that no matter how higher-order VPFs employed in practice, the period of spatial polarization of the VPF should be controlled as large enough relative to the size of the moving particle so as to guarantee the distinguishable DPS. Therefore, there should be a trade-off between the temporal resolution (mode order) and the relative size of VPF for effective real-time tracking of a moving particle.

## Discussion

In general, the scattering of light by a particle is a complex topic. It is worth noting that Mie scattering is unidirectional, becoming more so as the particle becomes larger[48,49], while omnidirectional for Rayleigh scattering. Thus photodetectors positioned along the line of sight would as a rule of thumb suffice for most objects, with due consideration for smaller particles. As the particle becomes larger, Mie scattering calculations for the single scatter become very close to geometric calculations[50]. In our proof of principle experiment, the moving particle is emulated by a lump of 78 adjacent micromirrors in the DMD[15,16], with a diameter of about 137 μm, approximating a geometric reflection since the size of micromirrors is two orders of magnitude larger than the wavelength of the light (see Methods). Thus, in our case, we could directionally reflect the local light of the VPF and collect a returned component. The photodetectors are aligned with the reflected light, and we envisage large particle tracking in real-world scenarios to mimic this. As for the loss, we had no problems with this using commercially available high-sensitive photodetectors, and for both a reflected and Mie scattered signal, the light becomes ever more unidirectional, so this is not an issue. In particular, the widely studied lidar system has been collecting scattered signals from many tens of km away, so achieving signal to noise in practical tracking situations could be feasible with the correct technology.

The vectorial Doppler effect with spatially variant polarized light fields enables the measurement of the rotational velocity vector (magnitude and direction). It also shows a potential to implement the real-time tracking of the instantaneous position of the moving particle under conditions of the known starting position and continuous tracking. This is because of the periodic polarization distribution along the azimuthal direction of the high-order VPFs, making the absolute position ambiguous. The precise knowledge of particle path and starting position limits the practical position tracking applications to a certain extent. One possible solution is to employ other spatially variant polarized light fields, such as light beams with a lemon/star type polarization singularity on its axis[51,52], corresponding to the rotation of the polarization vector by only π/2 after one cycle along the azimuthal direction. This allows for absolute position tracking as the instantaneous position of the moving particle can be determined by its corresponding unique local polarization along the azimuthal direction (rotational motion).

In addition to the lemon/star type VPFs, Berg-Johansen et al.[45] proposed another method to implement absolute position tracking. Classically entangled optical beams, i.e., low-order radially polarized beams, were employed for high-speed kinematic sensing with impressive performance[45]. It relies on the intrinsic correlations existing in vector optical beams between transverse spatial modes and polarization. The polarization was determined by Stokes parameters from intensity measurements with photodetectors. Remarkably, the optical beams were received by the photodetectors as a whole for measuring the "global polarization" and keeping the classical entanglement. The knowledge of the Stokes parameters of the "global polarization",

which was disturbed by the moving particle, allowed the object's instantaneous trajectory to be reconstructed. For comparison, the presented vectorial Doppler effect focuses more on the velocity vector (magnitude and direction) measurement using high-order VPFs. The principle relies on the locally sampled or reflected or scattered time-varying polarization light signal by the moving particle in the whole spatially variant polarized light field. It is the local scattering event and the photodetectors only receive the locally sampled light for measuring the "global polarization". It uses the FFT method to analyze the recorded time-varying polarization signals. The magnitude of the rotational velocity is deduced from the peak in the Fourier frequency spectra. Meanwhile, the direction of the rotational velocity is determined from the RPD between the detected intensity signals after two polarizers. Under the necessary conditions of knowing the particle path, starting position, and continuous tracking, absolute position tracking is achievable.

The abovementioned low-order lemon/star type VPFs[51,52] and radially polarized beams[45] can support absolute position tracking (uniqueness) but with less sensitivity when measuring the velocity. Remarkably, high-order cylindrical VPFs are advantageous to obtain high sensitivity of velocity measurement. This is because the measured frequency shift is proportional to the mode order of the cylindrical VPF. However, the knowledge of starting position and continuous tracking is needed to track the particle's instantaneous position. Meanwhile, for a given velocity range, high-order cylindrical VPFs might cause large frequency shifts exceeding the response bandwidth of photodetectors. Hence, there is a trade-off between uniqueness, velocity range, and sensitivity. For photodetectors with limited response bandwidth, appropriate high-order cylindrical VPFs should be considered to achieve a balance between the detected velocity range and sensitivity. In practical applications, even high-order cylindrical VPFs can work for a large detected velocity range when using fast enough photodetectors (e.g., tens of GHz). It is generally believed that low-order cylindrical VPFs are easier to be generated than high-order ones. In our experiment, we use a simple Sagnac interferometer configuration incorporating an SLM to generate cylindrical VPFs, which are formed by superposing two twisted-phase light beams (clockwise and counterclockwise in the Sagnac interferometer configuration). It is convenient to generate different order cylindrical VPFs simply by changing the pattern loaded onto the SLM. In particular, the Sagnac interferometer configuration sharing the same light path benefits the stable and high-quality generation of high-order cylindrical VPFs. In the experiment, we successfully generate $TM_{0,1}$, $HE_{2,1}$, $EH_{2,1}$, $HE_{4,1}$, $EH_{4,1}$, $HE_{6,1}$, $EH_{6,1}$, $HE_{8,1}$, $EH_{8,1}$, $HE_{10,1}$, and $HE_{19,1}$-like VPFs with high stability and beam quality (see Supplementary Fig. 5).

To measure a complicated motion beyond the rotation based on the vectorial Doppler effect, we might use three high-order cylindrical VPFs, each with a different wavelength, along $x$, $y$, and $z$ directions, respectively. Since the VPF along $x/y/z$ direction can be used to measure the velocity vector component along that direction (see Supplementary Note 2), for a complicated motion its velocity vector projections along $x$, $y$, and $z$ directions could be respectively measured assisted by wavelength filtering and demonstrated vectorial Doppler effect. Hence, it is possible to measure the velocity vector (magnitude and direction) information of a complicated motion. Meanwhile, under the necessary conditions of knowing the starting position and continuous tracking, it might be possible to further track the particle position of a complicated motion. Additionally, similar to the lemon/star type VPFs[51,52], we might fully utilize the degree of freedom of polarization tailoring and generate full Poincaré beams for instantaneous position tracking. We might use customized VPFs with different states of polarization not only along the azimuthal

direction but also along the radial direction (full Poincaré beams). Thus, any spatial position could be determined by its corresponding unique local polarization.

Remarkably, in the demonstrated proof of principle experiments, only isotropic particles are considered based on the assumption that a particle without birefringence does not change the polarization of the reflected/scattered light. This assumption is not rigorous for the more general case of anisotropic particles. For the interaction between the polarized light and the anisotropic particle, the polarization of the reflected/scattered light from a moving particle is dependent on the anisotropy of the particle such as its shape[53] and birefringence, as well as the spin of the particle around its center of mass. We use the Jones matrix method to study the anisotropic particle. The Jones matrix of a spinning anisotropic particle can be given as

$$
\begin{aligned}
\mathbf{M}(\Theta, t) &= \mathbf{R}(-\Theta t) \cdot \mathbf{M}_0 \cdot \mathbf{R}(\Theta t) \\
&= \begin{bmatrix} \cos \Theta t & -\sin \Theta t \\ \sin \Theta t & \cos \Theta t \end{bmatrix} \begin{bmatrix} m & 0 \\ 0 & n \cdot e^{i\delta} \end{bmatrix} \begin{bmatrix} \cos \Theta t & \sin \Theta t \\ -\sin \Theta t & \cos \Theta t \end{bmatrix}
\end{aligned}
\tag{5}
$$

where $\mathbf{R}(\Theta t)$ is the rotation matrix, $\Theta$ denotes the spin speed of the anisotropic particle, $m$ and $n$ are the real scatter/reflection coefficients along the short (fast) and long (slow) axes of the anisotropic particle, respectively, $\delta$ is the optical phase retardation between these two polarization components. When considering a moving anisotropic particle in both rotational motion and spinning motion (rotational velocity: $\Omega$, spin speed: $\Theta$) within the VPFs (see Supplementary Fig. 3), the reflected/scattered light by the moving particle, i.e., DPS with time-varying polarization, can be expressed as

$$
\begin{aligned}
\mathbf{E}(\Omega, \Theta, t) &\approx A \cdot \mathbf{M}(\Theta, t) \cdot \begin{bmatrix} \cos(\ell\Omega t - \alpha) \\ -\sigma \cdot \sin(\ell\Omega t - \alpha) \end{bmatrix} \\
&= \frac{nA}{2} \begin{bmatrix} (\tau + e^{i\delta}) \cdot \cos(\ell\Omega t - \alpha) + (\tau - e^{i\delta}) \cdot \cos[2\Theta t + \sigma(\ell\Omega t - \alpha)] \\ -\sigma(\tau + e^{i\delta}) \cdot \sin(\ell\Omega t - \alpha) + (\tau - e^{i\delta}) \cdot \sin[2\Theta t + \sigma(\ell\Omega t - \alpha)] \end{bmatrix}
\end{aligned}
\tag{6}
$$

where $\tau = m/n$. Especially, when $\tau = 1$ and $\delta = 0$, it is simplified to the case of the isotropic particle. It is worth noting that for an isotropic particle, whether it spins or not, there is no influence on the rotation measurement.

We then study the influence of anisotropy ($\tau$ and $\delta$) and spin ($\Theta$) of the moving particle on the rotation measurement by analyzing the Stokes parameters of the DPS in Eq. (6). We can use two groups of polarizers and photodetectors[45] to obtain the Stokes parameters ($S_0$, $S_1$, $S_2$, and $S_3$). For $S_1$ and $S_2$, there are three frequency components, $\Delta\omega_1 = 2\ell\Omega$, $\Delta\omega_2 = 2\Theta$, and $\Delta\omega_3 = 4\Theta + 2\sigma\ell\Omega$, where $\Delta\omega_1$ is caused by rotational motion only, while $\Delta\omega_2$ by spinning motion only. By properly switching the mode order of the cylindrical VPFs or based on the different relative amplitudes of three frequency components, we can distinguish the three frequency components and deduce the magnitude of rotational velocity and spin speed. Meanwhile, the direction of the rotational velocity and spin speed could be also determined through the RPD of the corresponding frequency component between $S_1$ and $S_2$ (see Supplementary Note 3).

Additionally, we might also use two polarizers and two photodetectors (similar approach used for isotropic particles) to analyze the DPS in Eq. (6). It is also possible to distinguish the rotation and spin of the anisotropic particle and measure the rotational velocity (magnitude and direction) and spin speed (magnitude and direction) (see Supplementary Note 3 and Supplementary Fig. 4). In a sense, the method of using two polarizers can be regarded as a simplified version of the method of analyzing Stokes parameters. The latter one measuring Stokes parameters gives the complete information of polarization (full Poincaré sphere), which may facilitate more polarization-related applications.

In summary, we propose and demonstrate the vectorial Doppler effect and the resultant vectorial Doppler metrology with the spatially variant polarized light fields, characterized by two-dimensional DPSs that can provide an additional RPD between different linear polarization components. Such RPD can be regarded as the Doppler phase shift, as an important supplementary of the conventional Doppler effect based on scalar optical fields only utilizing Doppler frequency shift. These unified Doppler frequency shift and Doppler phase shift allow for the full determination of the velocity vector (magnitude and direction) of a moving particle, i.e., vectorial Doppler metrology. In the proof of principle experiments, we successfully measure the rotational velocity and determine the direction of a moving isotropic particle. We also implement instantaneous position tracking of the moving particle with the knowledge of starting position and continuous tracking. The general case of anisotropic particles is also studied with Jones matrix method. It is possible to distinguish the rotation and spin of the anisotropic particle and measure the rotational velocity and spin speed (magnitude and direction) by analyzing Stokes parameters or using simplified two polarizers. It is remarkable that, naturally, this detection scenario might be further extended to complicated motion states by using three high-order cylindrical VPFs at different wavelengths or customized VPFs. This vectorial Doppler metrology approach enables the detection of the motion vector without using an additional reference light, and therefore shows high anti-interference to environmental disturbance. This benefits from the full exploitation of the spatially variant polarized light fields. Our findings may offer many emerging applications in Doppler velocimetry, metrology, and monitoring for universal motion vectors in the natural world and human industry.

## Methods

**Real-time position and velocity tracking**. Under the conditions of knowing the particle path, starting position, and continuous tracking, real-time tracking for instantaneous position and velocity of a particle with variable motion can be achieved by directly counting the peaks in the Doppler intensity spectra and meanwhile judging the Doppler phase shifts between two linearly polarized Doppler intensity signals. In general, for the spatially variant polarized light fields with a spatial period $\Lambda = \pi/k\gamma$ along the $x$ axis, one can track the instantaneous displacement of a moving particle relative to the initial position $x(t_0)$ as follows

$$
x(t_{m+1}) = \begin{cases} x(t_m) + \frac{\pi}{k\gamma}, & \Delta\varphi = -2\sigma\Delta\theta \\ x(t_m) - \frac{\pi}{k\gamma}, & \Delta\varphi = 2\sigma\Delta\theta \end{cases}
\tag{7}
$$

where $m = 0, 1, 2\ldots$ denotes the order number of peaks found in the Doppler intensity signals. $t_m$ corresponds to the moment of the $m^{\text{th}}$ intensity peak, and $t_0$ the starting time of counting peaks. The resulting instantaneous velocity vector at the moment of $t_{m+1}$ can be given as

$$
v(t_{m+1}) = \frac{v(t_{m+1}) - v(t_m)}{\Delta t_{m+1}} = \begin{cases} \frac{\pi}{\Delta t_{m+1}\gamma}, & \Delta\varphi = -2\sigma\Delta\theta \\ -\frac{\pi}{\Delta t_{m+1}\gamma}, & \Delta\varphi = 2\sigma\Delta\theta \end{cases}
\tag{8}
$$

where $\Delta t_{m+1} = t_{m+1} - t_m$ is the time interval between two adjacent intensity peaks.

For the real-time tracking of rotation movement of a particle using cylindrical VPFs in the experiment, the small rotational angle can be approximated as $d\phi = dx/r$, where $dx$ denotes the small displacement along the tangential direction and the skew angle $\gamma = \ell/kr$. Thus from Eq. (7), the instantaneous angular positions of the moving particle relative to its initial position $\phi(t_0)$ can be given as

$$
\phi(t_{m+1}) = \begin{cases} \phi(t_m) + \frac{\pi}{\ell}, & \Delta\varphi = -2\sigma\Delta\theta \\ \phi(t_m) - \frac{\pi}{\ell}, & \Delta\varphi = 2\sigma\Delta\theta \end{cases}
\tag{9}
$$

Similarly, the resulting rotational velocity can be given as

$$
\Omega(t_{m+1}) = \frac{\phi(t_{m+1}) - \phi(t_m)}{\Delta t_{m+1}} = \begin{cases} \frac{\pi}{\Delta t_{m+1}\ell}, & \Delta\varphi = -2\sigma\Delta\theta \\ -\frac{\pi}{\Delta t_{m+1}\ell}, & \Delta\varphi = 2\sigma\Delta\theta \end{cases}
\tag{10}
$$

**Generation of VPFs and mimicking of the moving particle**. We build an experimental setup for the detection of motion vector using the vectorial Doppler effect based on VPFs (see Supplementary Fig. 5). The generation of VPFs is incorporated into the experimental setup. A He–Ne laser beam at 632.8 nm is

shaped into a perfect Gaussian beam by propagating it through a piece of single-mode fiber (SMF). A group of polarizer (Pol.1) and half-wave plate (HWP1) is used to adjust the SoP of light beam to be $45^0$. When passing through a beam splitter (BS1) and a polarization beam splitter (PBS), the $45^0$ polarized light beam is spilt into $x$ and $y$-polarized components in two paths. The light beam in each path is modulated into a twisted-phase beam by an SLM in a Sagnac interferometer configuration. The HWP2 in the Sagnac interferometer configuration is used to rotate the linear SoP of light beams in both paths to its orthogonal state so that each light beam is modulated and then combined into a superposed beam through the PBS. Considering the reflection with odd and even times respectively for the two generated twisted-phase beams by the same SLM, the superposed beam contains two twisted-phase components with opposite topological charge number ($\pm\ell$) in $x/y$ linear SoP. After the BS and passing through a quarter-wave plate (QWP1), the superposed beam becomes the cylindrical VPF.

The Sagnac interferometer configuration sharing the same light path benefits the high-quality generation of cylindrical VPFs. In the experiment, cylindrical VPFs analogous to $HE_{41}$ vector mode and $EH_{21}$ vector mode (see Supplementary Fig. 5) are generated by imprinting with a computer-generated phase profile ($|\ell|=3$) to SLM and rotating QWP1 to control circular SoP. Other orders of cylindrical VPFs are also generated with high stability and beam quality using the Sagnac interferometer configuration (see Supplementary Fig. 5). The generated cylindrical VPF is illuminated on a moving particle that is mimicked through a DMD by setting micromirrors in specific time-varying positions to the state of "ON". We turn a lump of 78 adjacent micromirrors to the "ON" state to mimic a microcircle with a diameter of about 137 μm. The diameter of the cylindrical VPF is controlled to be about 2.5 mm to match the rotational radius of the moving particle. Otherwise the moving particle can interact with imprecise local SoP, making the DPS fuzzy and undistinguishable. In the experiment, the size ratio of the particle to the period of spatial SoP variation in VPF is about 0.1. The QWP2 is used for the compensation of the polarization-dependent dissipation when reflecting the local SoP by the mimicked particle consisting of an array of micromirrors. The apertures (AP1 and AP2) can block the light in undesired diffraction orders generated by SLM and DMD.

## Data availability

All the findings of this study are available in the main text or the Supplementary Information. The raw data are available from the corresponding author upon reasonable request.

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

## Acknowledgements

We would like to thank Prof. Alan E. Willner, Prof. Miles J. Padgett, Prof. Juan P. Torres, and Prof. Siddharth Ramachandran for their technical supports. This work was supported by the National Key R&D Program of China (2019YFB2203604), the National Natural Science Foundation of China (NSFC) (11774116), the Key R&D Program of Guangdong Province (2018B030325002), the Key R&D Program of Hubei Province of China (2020BAB001), and the Science and Technology Innovation Commission of Shenzhen (JCYJ20200109114018750).

## Author contributions

J.W. and L.F. developed the concept and conceived the experiments. Z.W. carried out the experiments and acquired the experimental data. L.F. performed the theoretical analyses. Z.W. and L.F. carried out the data analysis. L.F., A.F., and J.W. contributed to writing the paper. J.W. finalized the paper. J.W. supervised the project.

## Competing interests

The authors declare no competing interests.
