## [Peer Review File · Nature Communications]

REVIEWER COMMENTS

Reviewer #1 (Remarks to the Author):

Review of the manuscript submitted for publication to Nature Communications:

Vectorial Doppler Metrology,
by L. Fang, Z. Wan, A. Forbes, and J. Wang

The manuscript submitted can be of interested to a wide audience and this is why I think its publication in Nature Communications can be useful. There is no doubt that the development of methods to determine the velocity (magnitude and direction) of objects is important, especially from a practical point of view.

The manuscript presents a scheme that makes use of an optical field with a spatially-varying polarization to determine the magnitude and direction of the velocity of an object. The idea is interesting and introduce a new way for measuring velocities. The research area concerned with the generation and use of structured light, with both scalar and vector fields, has seen many important contributions during the last decade, fundamental and applied. This work can be one of these important contributions.

I would like to point out two remarks made by the authors. I would suggest that the authors reconsider, or reformulate, them:

- The manuscript says (see abstract): ``A consequence of the scalar wave approach is that it is not possible to deduce the motion direction of moving targets directly. Here we overcome this fundamental limitation....''.

I do not think there is a fundamental limit concerning the capacity of scalar waves to retrieve full information about the velocity of an object. There are schemes that aim at measuring the full velocity with scalar waves. For instance, one scheme considers using multiple laser beams, or a laser beam rapidly changing its direction of propagation, pointing at the object along different directions, and detecting Doppler shifts along different directions [see for instance Laser-Doppler measurement of crosswind velocity, *Appl. Opt.* 21, 2596–2607 (1982)]. Other schemes, with the help of a priori knowledge about the object under study and with the help of heavy computational efforts, can retrieve information about the full velocity. Other schemes, that the authors cite (see refs. 3-8), make use of scalar structured light beams to obtain full information of the velocity, not only its magnitude.

So scalar waves in smart scenarios can retrieve the full velocity of an object. Another issue is that these systems might be difficult to implement, provide low spatial resolution, be valid only in specific situations, require a priori information or being highly dependent on computational and numerical modelling. But these are technical limitations rather than fundamental. With this I do not mean that the present manuscript is less important. The use of structured polarized beams can be a tool to overcome the limitations of some of the methods discussed above.

- The manuscript says (see abstract): `` Strictly speaking, the vectorial Doppler effect does not shift the frequency of the employed wave, but modulates the polarization of its locally reflected/scattered light by the moving particle...In a broad sense, we may understand this extra modulated frequency as Doppler frequency shift...Consequently, despite its slight difference compared to the traditional linear Doppler effect, we also call it Doppler frequency shift...''.

With these sentences the authors seem to acknowledge that the use of the term "Doppler" for what they do can be confusing. Or maybe they try to widen the meaning of Doppler effect to refer to any time-varying signal whose temporal change is related to the movement of an object. I think this might be confusing for the reader. Actually, the title of the paper (Vectorial Doppler metrology) contains the word Doppler. In general, people refer to the Doppler effect as a phase change of a wave as a product of its interaction (reflection or scattering) with a moving object. This time varying phase modulation produces a frequency

shift that is the Doppler shift.

In summary, I think that this manuscript can be useful and interesting to a large audience, providing a new application of structured light beams. As a proof-of-concept of a new idea, I would recommend publication in Nat. Comm. How far in terms of real applications the scheme proposed can go would require further analysis.

The referee,
14th January 2021

Reviewer #2 (Remarks to the Author):

In this manuscript the authors use non-uniformly polarized beams of light, also known as “vector beams” or “structured light beams”, to perform a novel kind of Doppler metrology. The underlying concept is quite simple: they extend the standard scalar Doppler approach to vector beams, for measuring the magnitude and the direction of a (isotropic) particle undergoing a circular motion within the cross-section of the beam. This is basically achieved via a time Fourier transform of the polarization signal (AKA, Stokes parameters). Under more restrictive conditions, they are also able to track the instantaneous position of a particle, but this result was already obtained without restrictions by other authors. Finally, they also discuss the extension of their technique to anisotropic particles. It must be remarked that this is a “proof of principle” experiment because the moving particle does not really exist, it is mimicked by a digital micromirror device (DMD).

The article is clearly written and, in my opinion, presents enough novelty and general interest to deserve publication on the pages of Nature Communication. However, there is an important point that I could not understand neither from the manuscript, nor from the supplementary material. In the scheme employed by the authors, two laser beams meet at a paraxial angle γ to form a structured polarization/intensity pattern crossed by the particle. Then, the light scattered by the moving particle is measured by a suitably placed detector. Throughout the manuscript, a paraxial description of light is used. However, the light scattered by a Mie process typically does not present a strong directionality. So, I suppose that the detector is placed at a suitable chosen distance from the scattering center, so that only a paraxial pencil of light is selected and measured, otherwise the whole Stokes formalism would be meaningless. I think that the authors should add a brief discussion about this point, also mentioning the (possible) role of loss due to the scattered light selection.

Title: Vectorial Doppler Metrology

Author: Liang Fang, Zhenyu Wan, Andrew Forbes, and Jian Wang

Dear Reviewers of Nature Communications,

We would like to thank both reviewers for their positive comments on our work, the recommendation for publication, and the constructive comments to improve the work. We have revised our paper (Manuscript ID: NCOMMS-20-03487A-Z) based on the reviewers' comments and enclosed please find the detailed response.

Sincerely yours,

Liang Fang, Zhenyu Wan, Andrew Forbes, and Jian Wang

Response to the comments of two reviewers

Reviewer #1 (Remarks to the Author):

Review of the manuscript submitted for publication to Nature Communications:
Vectorial Doppler Metrology,
by L. Fang, Z. Wan, A. Forbes, and J. Wang

The manuscript submitted can be of interested to a wide audience and this is why I think its publication in Nature Communications can be useful. There is no doubt that the development of methods to determine the velocity (magnitude and direction) of objects is important, especially from a practical point of view.

The manuscript presents a scheme that makes use of an optical field with a spatially-varying polarization to determine the magnitude and direction of the velocity of an object. The idea is interesting and introduce a new way for measuring velocities. The research area concerned with the generation and use of structured light, with both scalar and vector fields, has seen many important contributions during the last decade, fundamental and applied. This work can be one of these important contributions.

Reply:

The authors are very grateful to the reviewer for giving very positive comments on our work.

I would like to point out two remarks made by the authors. I would suggest that the authors reconsider, or reformulate, them:

- The manuscript says (see abstract): "A consequence of the scalar wave approach is that it is not possible to deduce the motion direction of moving targets directly. Here we overcome this fundamental limitation....".

I do not think there is a fundamental limit concerning the capacity of scalar waves to retrieve full information about the velocity of an object. There are schemes that aim at measuring the full velocity with scalar waves. For instance, one scheme considers using multiple laser beams, or a laser beam rapidly changing its direction of propagation, pointing at the object along different directions, and detecting Doppler shifts along different directions [see for instance Laser-Doppler measurement of crosswind velocity, Appl. Opt. 21, 2596–2607 (1982)]. Other schemes, with the help of a priori knowledge about the object under study and with the help of heavy computational efforts, can retrieve information about the full velocity. Other schemes, that the authors cite (see refs. 3-8), make use of scalar structured light beams to obtain full information of the velocity, not only its magnitude.

So scalar waves in smart scenarios can retrieve the full velocity of an object. Another issue is that these systems might be difficult to implement, provide low spatial resolution, be valid only in specific situations, require a priori information or being highly dependent on

computational and numerical modelling. But these are technical limitations rather than fundamental. With this I do not mean that the present manuscript is less important. The use of structured polarized beams can be a tool to overcome the limitations of some of the methods discussed above.

Reply:

The authors thank the reviewer for pointing out this loose expression. The reviewer is correct. The statements "...it is not possible..." and "...overcome this fundamental limitation" are not rigorous. Indeed, if one allows multiple laser beams or a laser beam rapidly changing its direction of propagation (see [1] Laser-Doppler measurement of crosswind velocity, *Appl. Opt.* 21, 2596-2607 (1982)), it is possible to measure the full velocity with scalar waves. Other schemes with the help of a priori knowledge about the object under study and heavy computational efforts are also applicable. As pointed out by the reviewer, scalar waves in smart scenarios may retrieve the full velocity of an object, but come at the cost of difficulty in implementation, high complexity, or highly dependent on computational modelling. It is technically challenging rather than fundamental. Our approach overcomes this challenge with just a single vectorially structured beam.

Following the reviewer's valuable suggestions, in the revised paper (see *Abstract* and *Introduction* sections), we rewrite the inappropriate expressions and cite the helpful reference provided by the reviewer. For example,

"A consequence of the scalar wave approach is that it is not possible to deduce the motion direction of moving targets directly. Here we overcome this fundamental limitation...."

is changed to

"A consequence of the scalar wave approach is that it is technically challenging to directly deduce the motion direction of moving targets, ... Here we overcome this challenge...".

References:

[1] Durst, F., Howe, B. M. & Richter, G. Laser-Doppler measurement of crosswind velocity. *Appl. Opt.* 21, 2596-2607 (1982).

(Please see pages 1, 2 and Ref. [22] in the revision)

- The manuscript says (see abstract): "Strictly speaking, the vectorial Doppler effect does not shift the frequency of the employed wave, but modulates the polarization of its locally reflected/scattered light by the moving particle...In a broad sense, we may understand this extra modulated frequency as Doppler frequency shift...Consequently, despite its slight difference compared to the traditional linear Doppler effect, we also call it Doppler frequency shift...".

With these sentences the authors seem to acknowledge that the use of the term "Doppler" for what they do can be confusing. Or maybe they try to widen the meaning of Doppler effect to refer to any time-varying signal whose temporal change is related to the movement of an object. I think this might be confusing for the reader. Actually, the title of the paper (Vectorial Doppler metrology) contains the word Doppler. In general, people refer to the Doppler effect as a phase change of a wave as a product of its interaction (reflection or scattering) with a moving object. This time varying phase modulation produces a frequency shift that is the Doppler shift.

Reply:

The authors thank the reviewer for bringing this to our attention. The reviewer is correct. Our aim here is indeed to widen the meaning of the Doppler effect to vectorial light, and with this highlight the benefits to be derived.

Following the reviewer's valuable suggestions, in the revised paper, we rewrite these sentences to avoid confusion for readers.

(Please see page 6 in the revision)

In summary, I think that this manuscript can be useful and interesting to a large audience, providing a new application of structured light beams. As a proof-of-concept of a new idea, I would recommend publication in Nat. Comm. How far in terms of real applications the scheme proposed can go would require further analysis.

Reply:

The authors thank the reviewer for the very positive comments and recommendation on publication of our work in Nature Communications. We improve the work by rewriting the inappropriate statements according to the reviewer's constructive comments. We also agree with the reviewer that further analysis would be required towards real applications.

(Please see pages 1,2,6 in the revision)

Reviewer #2 (Remarks to the Author):

In this manuscript the authors use non-uniformly polarized beams of light, also known as "vector beams" or "structured light beams", to perform a novel kind of Doppler metrology. The underlying concept is quite simple: they extend the standard scalar Doppler approach to vector beams, for measuring the magnitude and the direction of a (isotropic) particle undergoing a circular motion within the cross-section of the beam. This is basically achieved via a time Fourier transform of the polarization signal (AKA, Stokes parameters). Under more restrictive conditions, they are also able to track the instantaneous position of a particle, but this result was already obtained without restrictions by other authors. Finally, they also discuss the extension of their technique to anisotropic particles. It must be remarked that this is a "proof of principle" experiment because the moving particle does not really exist, it is mimicked by a digital micromirror device (DMD).

Reply:

We would like to thank the reviewer for the good summary of our work and the overall positive comments. The reviewer is correct. We agree that our approach is proof of principle, which we implemented using a DMD for convenience and so that we had control over the current available experimental conditions in the laboratory. Actually, mimicking the moving particle with a DMD has also been widely reported in previous works, which could be helpful to wide readers (see [1][2] Rosales-Guzmán, 2013, 2014).

Following the reviewer's valuable comments, in the revised paper (see *Abstract, Introduction, Experimental Results, Discussion* and *Conclusion* sessions), we clearly remark that it is a "proof of principle" experiment.

References:

[1] Rosales-Guzmán, C., Hermosa, N., Belmonte, A. & Torres, J. P. Experimental detection of transverse particle movement with structured light. *Sci. Rep.* **3**, 2815 (2013).

[2] Rosales-Guzmán, C., Hermosa, N., Belmonte, A. & Torres, J. P. Measuring the translational and rotational velocities of particles in helical motion using structured light. *Opt. Express* **22**(13), 16504-16509 (2014).

(Please see pages 1,3,7,11,13,15 and Refs. [15][16] in the revision)

The article is clearly written and, in my opinion, presents enough novelty and general interest to deserve publication on the pages of Nature Communication. However, there is an important point that I could not understand neither from the manuscript, nor from the supplementary material. In the scheme employed by the authors, two laser beams met at a paraxial angle γ to form a structured polarization/intensity pattern crossed by the particle. Then, the light scattered by the moving particle is measured by a suitably placed detector. Throughout the manuscript, a paraxial description of light is used. However, the light scattered by a Mie process typically does not present a strong directionality. So, I suppose that the detector is placed at a suitable chosen distance from the scattering center, so that only a paraxial pencil of light is selected and measured, otherwise the whole Stokes

formalism would be meaningless. I think that the authors should add a brief discussion about this point, also mentioning the (possible) role of loss due to the scattered light selection.

Reply:

We would like to thank the reviewer for the positive comments on the novelty and general interest of our work, as well as the recommendation for publication on Nature Communications. In particular, we thank the reviewer for raising the point about the light scattered by the moving particle, and we fully agree with the reviewer that it deserves clarification. Following the reviewer's valuable suggestions, in the revised paper, we add a brief discussion about this point and also mention the role of loss due to the scattered light selection.

Scattering of light is a complex topic. As the reviewer correctly points out, for small collection angles and "far field" measurement, the transport factor in the relation between the incident and scattered light field becomes ~ 1 , while the scattering matrix becomes diagonal (see [1] Hansen and Travis, 1974) for both Mie and Rayleigh scattering. It should be noted that Mie scattering is actually uni-directional, becoming more so as the particle becomes larger, while omni-directional for Rayleigh scattering. Thus a detector positioned along the line of sight would as a rule of thumb suffice for most objects, with due consideration for smaller particles. As the particle becomes larger, Mie scattering calculations for the single scatter become very close to geometric calculations (see [2] Born and Wolf, 1959). In our proof of principle experiment, the particle was emulated by micromirrors in the DMD, with a diameter of about 137 μm , similar to some other previous works (see [3][4] Rosales-Guzmán, 2013, 2014), thus approximating a geometric reflection since the size of micromirrors is two orders of magnitude larger than the wavelength of the light. Thus, in our case we could directionally reflect the local light of the vectorial polarization field and collect a returned component. Our detectors were aligned with the reflected light, and we envisage large particle tracking in real-world scenarios to mimic this. One can imagine that in principle any detection position might be possible if the scattering matrix could be inverted, but this is too speculative at the moment to suggest (but worth looking into all the same). As for the loss, we had no problems with this using commercially available detectors, and for both a reflected and Mie scattered signal, the light becomes ever more uni-directional, so this is not an issue. Having said that, the LIDAR community has been collecting scattered signals from many tens of km away, so achieving signal to noise in practical tracking situations (surely not 50 km away) should be feasible with the correct technology. Just a small clarification: our approach actually uses one single vectorially structured beam, which is also one of the benefits of our scheme. The angle γ in supplementary materials (Fig. S1a), in relation to the detector, is for the simple conceptual illustration from the traditional point of view.

The above discussions and references are properly added in the revised paper.

References:

- [1] Hansen, J. E. & Travis, L. D. Light scattering in planetary atmospheres. *Space Sci. Rev.* **16**, 527-610 (1974).
- [2] Born, M. & Wolf, E. *Principles of Optics*. Pergamon Press, New York (1959).
- [3] Rosales-Guzmán, C., Hermosa, N., Belmonte, A. & Torres, J. P. Experimental detection of transverse particle movement with structured light. *Sci. Rep.* **3**, 2815 (2013).
- [4] Rosales-Guzmán, C., Hermosa, N., Belmonte, A. & Torres, J. P. Measuring the translational and rotational velocities of particles in helical motion using structured light. *Opt. Express* **22**(13), 16504-16509 (2014).

(Please see page 11 and Refs. [15][16][49][50] in the revision)

Thanks again for the valuable comments and helpful suggestions.

REVIEWERS' COMMENTS

Reviewer #1 (Remarks to the Author):

Dear editor of Nat. Comm., and authors of the paper,

In my first review of the manuscript I recommended its publication. I also suggested to make a few changes to the manuscript in order to clarify a few points. The authors has changed the manuscript accordingly so I keep my recommendation for publishing the manuscript. I think that what the authors do is this manuscript important and of general interest. Some statements that were slightly confusing in the previous version of the manuscript has been changed and are now clear.

The referee.

Reviewer #2 (Remarks to the Author):

I think that the authors have replied properly to all comments by the two referees, and amended consequently the manuscript. I believe that it is now suitable for publication in Nature Communications.

Title: Vectorial Doppler Metrology

Author: Liang Fang, Zhenyu Wan, Andrew Forbes, and Jian Wang

Response to the comments of two reviewers

REVIEWERS' COMMENTS

Reviewer #1 (Remarks to the Author):

Dear editor of Nat. Comm., and authors of the paper,

In my first review of the manuscript I recommended its publication. I also suggested to make a few changes to the manuscript in order to clarify a few points. The authors has changed the manuscript accordingly so I keep my recommendation for publishing the manuscript. I think that what the authors do is this manuscript important and of general interest. Some statements that were slightly confusing in the previous version of the manuscript has been changed and are now clear.

The referee.

Reply:

The authors thank the reviewer for the positive comments and recommendation for publication in Nature Communications.

Reviewer #2 (Remarks to the Author):

I think that the authors have replied properly to all comments by the two referees, and amended consequently the manuscript. I believe that it is now suitable for publication in Nature Communications.

Reply:

The authors thank the reviewer for the positive comments and recommendation for publication in Nature Communications.